# Spiritual Distress, Hopelessness, and Depression in Palliative Care: Simultaneous Concept Analysis

**DOI:** 10.3390/healthcare12100960

**Published:** 2024-05-07

**Authors:** Helga Martins, Rita S. Silva, Joana Bragança, Joana Romeiro, Sílvia Caldeira

**Affiliations:** 1Postdoctoral Program in Integral Human Development, Católica Doctoral School, 1649-023 Lisbon, Portugal; jromeiro@ucp.pt; 2Health School, Polytechnic Institute of Beja, 7800-000 Beja, Portugal; 3Centre for Interdisciplinary Research in Health, Universidade Católica Portuguesa, 1649-023 Lisbon, Portugal; scaldeira@ucp.pt; 4Faculty of Health Sciences and Nursing, Universidade Católica Portuguesa, 1649-023 Lisbon, Portugal; s-arcssilva@ucp.pt (R.S.S.); s-jfbraganca@ucp.pt (J.B.)

**Keywords:** caregivers, depression, Haase’s model, hopelessness, palliative care, Rodger’s evolutionary model, simultaneous concept analysis, spiritual distress

## Abstract

Spiritual distress, hopelessness, and depression are concepts that are often used in palliative care. A simultaneous concept analysis (SCA) of these concepts is needed to clarify the terminology used in palliative care. Therefore, the aim of this study is to conduct a SCA of spiritual distress, hopelessness, and depression in palliative care. A SCA was performed using the methodology of Haase’s model. A literature search was conducted in March 2020 and updated in April 2022 and April 2024. The search was performed on the following online databases: CINAHL with Full-Text, MEDLINE with Full-Text, MedicLatina, LILACS, SciELO, and PubMed. The search was achieved without restrictions on the date of publication. A total of 84 articles were included in this study. The results highlight that the three concepts are different but also share some overlapping points. Spiritual distress is embedded in the rupture of their spiritual/religious belief systems, a lack of meaning in life, and existential issues. Hopelessness is a sense of giving up and an inability to control and fix the patient’s situation. Finally, depression is a state of sadness with a multi-impaired situation. In conclusion, refining the three concepts in palliative care is essential since it promotes clarification and enhances knowledge development towards intervention.

## 1. Introduction

Palliative care (PC), while still an integrated component of modern healthcare, is increasingly recognized as an essential part of all healthcare systems and should be provided at all levels of care [1,2,3]. PC is the active, holistic care of individuals of all ages with profound health-related suffering from severe illness, especially those near the end of life [4]. The global burden of severe health-related suffering will almost double by 2060 [5]. Suffering is health-related when it is associated with illness or injury. Suffering is severe when it cannot be relieved without medical intervention and when it compromises physical, social, or emotional functioning. Palliative care should be focused on relieving the health-related suffering associated with life-limiting or life-threatening conditions or the end of life [6,7]. The integration of PC into universal health coverage is essential to mitigate the catastrophic weakening of health systems and to alleviate the suffering of millions of patients and their families [5]. PC aims to improve the quality of life of patients, their families, and their caregivers [4].

Family caregivers are still the most excellent source of support for patients in end-of-life care, especially when they wish to be cared for at home. Carers can be considered the core structure for the continuity of care of these patients [8].

There are several definitions of carers. The National Institute for Health and Clinical Excellence (UK) offers the following definition: ‘carers, who may or may not be family members, are lay people in a close, supportive role who share in the illness experience of the patient and who undertake vital care work and emotion management’ [9].

Caring in the final phase of life raises some specific issues, and carers have specific needs: psychological and emotional support, information, help with the personal, nursing, and medical care of the patient, out-of-hours and night support, respite, and financial help. A carer’s ability to cope reflects their individual circumstances and resources and may change throughout the patient’s illness [10].

Health professionals frequently underrate caregivers’ needs for information regarding palliative care, death, and dying, and may feel inadequately prepared to discuss these issues [11]. Caregivers simultaneously report their need for detailed information about what to expect as their loved one dies and difficulties in comprehending and receiving lousy news [11].

Although universally recognized, the role of the family caregiver still needs to be better supported by society, health teams, and family systems. Substantial knowledge about the actions performed by caregivers can facilitate good clinical and psychosocial practices. For instance, in the particular context of palliative care, the family simultaneously provides and receives healthcare. For the well-being of the patients, the caregivers perform different activities added to those they have in their routine, which is essential for the continuity of care of the patient at the end of their life, both at the hospital and at home [8].

Family caregivers experience limited involvement in planning palliative care. Their voices seem to be silenced, and the involvement of family caregivers is not in proportion to their responsibilities. Family caregivers’ involvement in palliative care should be an emergent topic in primary nursing education and professional education for nurses [12].

As mentioned above, palliative care is a fundamental part of health care in which several nursing diagnoses emerge in a negative light since it is a domain that is surrounded by suffering and existential issues. Thus, concepts of spiritual distress, hopelessness, and depression may be experienced by the patients [13,14].

Emphasizing the attention to the preceding concepts, they are included in nursing taxonomies and clinical practice. In 1978, NANDA-International, Inc. (NANDA-I) (Oconto Falls, WI, USA) included the nursing diagnosis of spiritual distress in the taxonomy. This diagnosis was reviewed in 2002 and 2013 [15]. Currently, spiritual distress arises at certain times in life in response to a serious health situation, and the patients are involved in the sense of lack of meaning in life and suffering [16].

On the other hand, hopelessness is a nursing diagnosis according to NANDA-I, and is described as a subjective state where the individuals have limited or no choices on their behalf [14]. Finally, depression is not a nursing diagnosis in the NANDA-I taxonomy, and is considered a related factor, risk factor, or a defining characteristic [14]. However, in the International Classification for Nursing Practice (ICNP), there is a diagnosis of depressed mood, which embraces a negative feeling of sadness or melancholy [17].

These concepts have many similarities and are interconnected, and a distinction between the three concepts seems difficult to achieve in clinical nursing practice. The latter statement is supported by Haase et al. [18]; there are certain concepts that are difficult to define, bringing additional difficulties that may cause theoretical obstacles and references to these as distinct concepts. Hence, it is essential to analyze the concepts of spiritual distress, hopelessness, and depression to clarify the terminology used regarding people with palliative care needs and caregivers in clinical practice. Additionally, clarifying spiritual distress, hopelessness, and depression would be helpful for clinical reasoning and in planning nurses’ interventions in clinical practice. The purpose of this study is to conduct a simultaneous concept analysis to define the similarities and particularities of spiritual distress, hopelessness, and depression in people with palliative care needs and caregivers, as described in the nursing literature. In addition, this SCA will achieve a theoretical definition of the concepts of spiritual distress, hopelessness, and depression to refine and clarify the concepts, facilitating clinical reason in nurses’ clinical practice.

## 2. Materials and Methods

The research question:What is the definition of spiritual distress, of hopelessness, and of depression in people with palliative care needs and caregivers?

This study used the nine-step approach of SCA by Haase et al. [18] based on Rodger’s evolutionary model. The SCA method is a “strategy designed to analyze interrelationships and identify theoretical overlap, common themes, and distinguishing characteristics among similar or complementary concepts” [18] (p. 227).

Next, the nine steps of the SCA [17] will be described in detail (Figure 1):

Data source/Literature review

A search conducted in international online databases was performed via EBSCOHost (CINAHL with Full-Text; MEDLINE with Full-Text; MedicLatina), LILACS, SciELO, and PubMed. The search was performed on March 2020 and updated on April 2022 and April 2024, without restrictions on the date of publication. No time limit was established to allow a greater range of results.

The search of each database followed a personalized algorithm search for each concept analyzed in this study (detailed information is summarized in Appendix A) in the title/abstract. Results were imported into the reference management software EndNote X8.

Inclusion criteria: original papers concerning spiritual distress or hopelessness, or depression concerning patients or caregivers in the context of palliative care; original papers [regardless of the research method], thesis, editorials, discussion, or opinion pieces. Studies in English, French, Spanish, and Portuguese were considered. Search, screening, extraction, and data analysis were performed by two independent reviewers [H.M. and R.S.S.], and two reviewers were included to solve disagreements [J.B. and J.R.]. The entire process was supervised by a senior researcher [S.C.].

In the beginning, 3232 citations were included. After deleting the duplicates with EndNote X8, 1460 articles were saved. After reading the title/abstracts, 277 articles were kept. The full-text review yielded 84 articles for inclusion. The selection process is summarized in Table 1. The full details are available in the Appendix A.

**Figure 1 healthcare-12-00960-f001:**
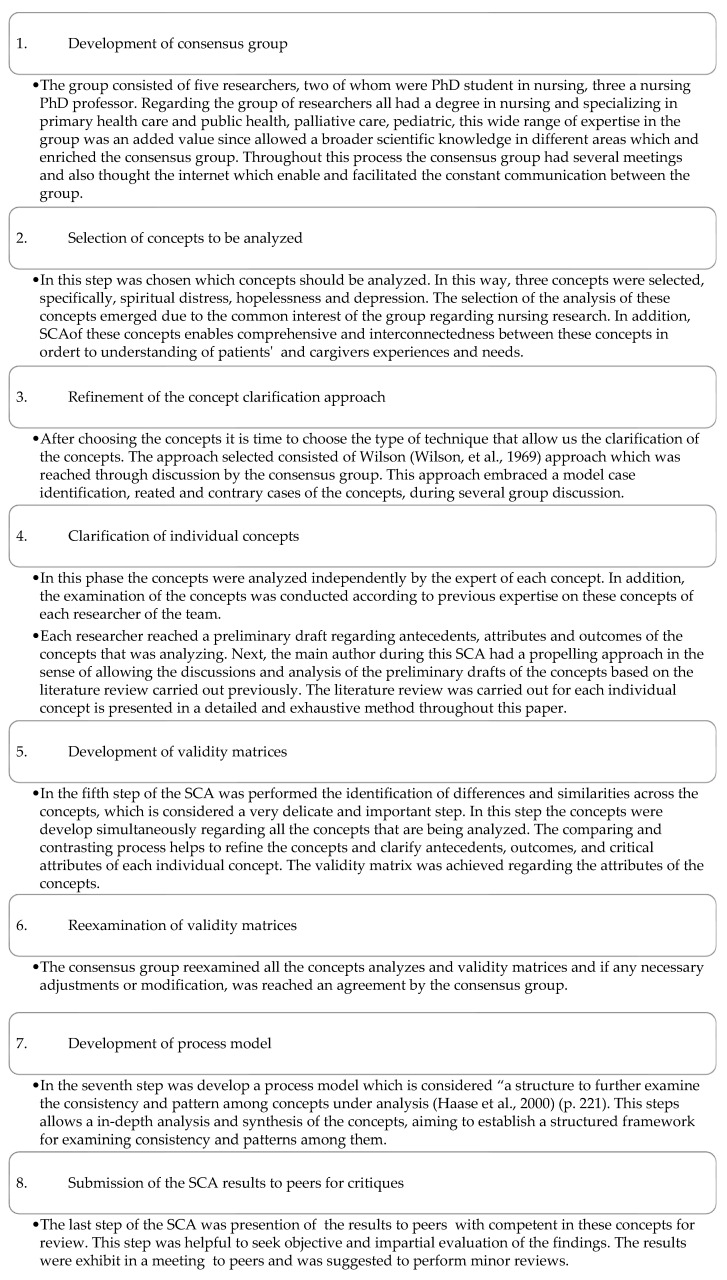
SCA process [18,19].

## 3. Results

The results are presented individually to allow a full understanding of the unique and individual approach to each concept. Further information is available in the Appendix A.

### 3.1. Spiritual Distress

The proposed conceptual definition of spiritual distress is a disconnection from self, others, and God or the transcendent. Individuals express a lack of meaning and purpose in life, which is due to the rupture of their spiritual/religious belief system.

The surrogate terms of spiritual distress that have been identified were existential distress, existential suffering, spiritual suffering, spiritual pain, and spiritual struggle.

Enablers regarding spiritual distress are lack of knowledge concerning spiritual distress, difficulty in distinguishing spiritual and religious needs, lack of time, lack of education, personal topic, and organization issues.

The obstacles are comfort and compassionate care, social and family support, a healing environment, spiritual support, and an interprofessional and holistic approach to patients. Table 2 presents the process model of spiritual distress.

### 3.2. Hopelessness

There are several surrogate terms for the concept of hopelessness, such as depression, despair, helplessness, pessimism, powerlessness, and worthlessness.

On one hand, the enablers are the absence of firm beliefs or strong faith, denial, and lack of pain control. On the other hand, the obstacles are spirituality, resilience, and acceptance.

The definition of hopelessness in this study is an emotional condition in which the individuals express a sense of giving up, an inability to improve and control one´s situation, leading to mental health decline and the desire for a hastened death. Detailed information concerning the process model of hopelessness is available in Table 3.

### 3.3. Depression

The main obstacles are spirituality, acceptance, and resilience. The enablers are sleep disorders, anxiety, negative social interactions, women, and people with a low education level. Surrogate terms regarding depression are depressed mood, depressive disorders, mood disorders, and sadness.

The definition proposed, concerning depression is a concept that emerges from emotional exhaustion and psychological distress, in which the individual expresses sadness and has multi-impaired function, possibly leading to a negative overall impact on patients’ lives and increased mortality. For a visual representation of the process model of depression, Table 4 is presented.

### 3.4. Validity Matrix for Critical Attributes

The validity matrix for critical attributes was re-examined by all researchers. This is an important tool to assess the interrelationships and differences between the concepts but above all to achieve theoretical cogency [18]. Likewise, the validation matrices provide a vital approach because we can examine all the elements of the concepts in factors [18].

The validation matrix of spiritual distress, hopelessness, and depression allows us to expose the factors that make these concepts different from each other. In this way, we can see that the characteristic lack of meaning and purpose in life is a determinant for spiritual distress, while the hopelessness results show that there is a giving up in life. In turn, depression is characterized by multiple impairments in several domains. Spiritual distress presents a unique characteristic, which is a disconnection from God and a feeling of abandonment and punishment. For spiritual distress, the results show a disintegration of self; however, for depression, there is an impaired self-esteem, hopelessness, and the inability of the self to control or to improve their situation.

Further information is available in Table 5.

## 4. Discussion

The data yielded by this study provided clarifying and relevant evidence regarding the concepts of spiritual distress, hopelessness, and depression. This research determined the antecedents, attributes, consequences, and a new definition of the three concepts. This SCA is important in nursing since the clarification of concepts permits the classification or characterization of phenomena and the evaluation of the strengths and limitations of each concept. However, the concepts are known to be dynamic and change through time and with context [20]. Moreover, we want to highlight that concept analysis plays a significant role as it allows us to know the current state of science, and it is essential in developing nursing knowledge and quality of care [21,22,23].

The three concepts under study emerge from similar antecedents such as the caregivers’ burdens, and impaired relationship between the caregiver and patient, lack of financial and social support, and the deterioration of the patients’ health. The caregiver’s burden is due to the caregiving tasks and care demands [24], which are aggravated by the greater severity of the symptoms presented by the patient as the control and management of these symptoms becomes crucial [25]. In addition, one of the antecedents of our study is the lack of control of pain and lack of symptom management. These results are in agreement with a study by Eagar et al. [26], in which they report that about 26% of the patients at the beginning of palliative care have poor symptom management; however, this value decreases to 13.9% in the final stage of life. Although they are similar antecedents in this SCA, it was possible to distinguish different antecedents and attributes.

To begin with, spiritual distress emerges from the breakdown of spiritual/religious belief systems and the disintegration of the self, in which existential issues take center stage. These findings are only restricted to spiritual distress since the new definition proposed in this SCA indicates that depression and hopelessness are incompatible with the rupture of the spiritual/religious belief system of the patients and caregivers. A review performed by Martins and Caldeira [27] that focused on the patients’ experiences regarding spiritual distress also emphasized that spiritual distress occurs when patients have a rupture of their spiritual and religious beliefs and have existential issues. According to Roze des Ordons et al. [28] (p. 129), these existential issues are associated with “meaning, identity, autonomy, dignity, support, connectedness, relationships, stress, anxiety, guilt, isolation, hope, fear, and anxiety”. Based upon these existential characteristics, we also found similarities in the spiritual distress concept.

The spiritual distress concept must be updated as the latest concept analysis was performed by Villagomeza [29] almost 20 years ago, which embraced fundamental expressions of impairments of spirituality, such as connectedness, faith, religious belief systems, value systems, meaning and purpose in life, self-transcendence, inner peace and harmony, and inner strength and energy. Although it has been a long time since then, the concept of Villagomeza [29] could be accepted nowadays since the results are in line with the attributes of our study.

Depression happens in the context of psychological distress, stress, and emotional exhaustion. The prior studies in this area also underline that a stressful event is necessary for depression [30,31]. According to the vision of Wilson [31], a depressed mood and the loss of interest or pleasure in activities are attributes that are necessary to reach the diagnosis of depression. The results of this study show that a depressive mood is a surrogate term and not an attribute. In addition, ICPN considers the expression of a depressive mood with regard to the nursing diagnosis. The most recent concept analysis of depression states that depression is a complex concept related to sadness [30]; in our study, depression assumes a multi-impairment role regarding self-esteem, emotional processing, cognitive functioning, and social functioning, which supports the previous author’s understanding.

Regarding the concept of hopelessness, in the SCA, it undertakes the role of an attribute in the concept of depression and an outcome in the concept of spiritual distress. The previous study also found that hopelessness was an attribute of depression [30]. These discrepancies are not novel since questions about hopelessness are a reality in nursing (since this is a nursing diagnosis in both the ICNP and the NANDA-I) but can also be an outcome. Further studies regarding the role of hopelessness are needed since disclosures regarding hopelessness are vital in clinical practice in a palliative care setting to facilitate nurses’ clinical reasoning.

Our findings revealed that according to the identified attributes, hopelessness is a sense of giving up, and patients are unable to control their situation, which, compared to the concept of depression and spiritual distress, has an inherently more negative definition. The first concept analysis mentions that hopelessness is an endpoint for the patients and is grounded in a pessimism perspective [32]. The latest concept analysis mentions that hopelessness is a psychological response to a negative incident in a patient’s life [33]. In addition, there are negative expectations, thoughts, and feelings toward changing one´s future [33]. As shown by the results of our concept analysis, our results are in alignment with the previous study. Nevertheless, our SCA brings a new dimension to the sense of futility of life experienced in palliative care.

Another attribute considered recurrently in this SCA was guilt, which was present in the depression and spiritual distress concepts. Leget [34] endorses the idea that palliative care is necessary to promote forgiveness and reconciliation. Nevertheless, to achieve this goal, it is necessary for the healthcare team to take an interdisciplinary approach. The benefits of forgiveness in palliative care are well documented and are enormous, particularly regarding positive outcomes in the patient’s general health [34,35].

Regarding the enablers of spiritual distress in palliative care that were highlighted in this analysis of concepts, these are in line with the study by Bar-Sela et al. [36], which identified the barriers to spiritual care. Therefore, our study shows that there is a lack of operationalized spiritual care in palliative care, which leads to the promotion of spiritual distress. It is about time to change this paradigm and encourage the inclusion of the spiritual dimension in care since the benefit of palliative care is mainly manageable discomfort, giving hope and meaning in their lives [36]. In addition, the inclusion of spirituality in clinical practice enhances the quality of life for patients in palliative care [37].

The consequences from the three concepts analyzed all have a negative impact on patients and palliative care providers. Special attention to hopelessness and depression should be given since patients are at risk of suicidal ideation. This disturbing result is in alignment with the study of Ribeiro et al. [38].

The main limitation of the study concerns the language in the inclusion criteria, as results are published in other languages that reviewers need to be proficient in. Additionally, a quality assessment of the included papers was not conducted since there is a variety of empirical and theoretical studies included in this SCA.

Further studies determining the significant statistical relationships between the three concepts are advised. In addition, testing the efficacy and effectiveness of interventions that address these concepts will generate robust evidence to guide clinical decision-making in clinical practice. Furthermore, studies are recommended to improve the diagnostic accuracy of these concepts.

## 5. Conclusions

From this SCA, we can realize that there are interrelationships and overlaps between the three concepts. However, there are attributes that characterize the concepts studied individually. Regarding the scientific development of nursing, the contribution of this research created the clarification and refinement of the theoretical definition of concepts, providing better clinical reasoning for nurses and indorsing an improvement in diagnostic accuracy in clinical practice with a greater focus on palliative patients and their caregivers.

The implications for clinical practice in palliative care stemming from this simultaneous concept analysis are profound. By elucidating the interrelationships and overlaps between various concepts, this research facilitates a deeper understanding of the complex dynamics involved in caring for palliative patients and their caregivers.

## Figures and Tables

**Table 1 healthcare-12-00960-t001:** Number of citations of spiritual distress, hopelessness, and depression.

	CINAHL	MEDLINE	PubMed	SciELO	MedicLatina	LILACS	Total	Included Articles
Spiritual distress	450	612	668	0	5	0	1735	25
Hopelessness	27	36	52	0	3	0	118	18
Depression	339	502	536	0	2	0	1379	41

**Table 2 healthcare-12-00960-t002:** Process model: Spiritual distress.

Item	Characteristics
Antecedents	Awareness of terminality and death Caregiver’s burden Existential issues Lack of financial support Lack of social support Loss of autonomy Loss of control Loss of relationships Rupture of belief system or person’s spiritual/religious orienting system and/or their beliefs Sense of disintegration of the self Traumatic life events Uncertainty about future Unmet spiritual needs
Attributes	Alienation Disconnection from self, from others, and from God or the transcendent Existential issues, frequents thought about death Fear for the future Feeling abandoned by God, loss of faith, and/or religious/spiritual belief Feel anger and punished by God Guilt Inability to self-forgive Isolation Loneliness Loss of meaning and purpose in life Loss or altered sense of self Not feeling at peace Questioning the meaning of their experiences Suffering
Outcomes	Anxiety Decrease in quality of life Decrease in spiritual well-being Decrease in general well-being Denial Depression Hopelessness More behavioral disengagement More dysfunctional coping strategies More prone to severe/increasing pain

**Table 3 healthcare-12-00960-t003:** Process model: Hopelessness.

Item	Characteristics
Antecedents	Caregiver burden and sense of being a burden by the patient Impaired relationship between caregiver and patient Impaired medical curative treatment Perception of an incurable/life-threatening illness Perception of a negative health condition Physical and psychological deterioration Social isolation
Attributes	Having given up on life Inability to improve and control one’s situation Lack of future expectations Lack of hope Negative expectations about the future Negative feelings towards the future Negative thoughts and feelings Sense of futility of life Uncertainty regarding the future
Outcomes	Depression Despair Fatality Hastened death Impaired quality of life Suicidal ideation

**Table 4 healthcare-12-00960-t004:** Process model: Depression.

Item	Characteristics
Antecedents	Caregiver burden and sense of being a burden by the patient Deterioration of health of the patient Emotional exhaustion Impaired relationship between the caregiver and patient Lack of financial support Lack of social support Perception of an incurable/life-threatening illness Poor control of pain and symptoms Psychological distress Stress
Attributes	Guilt Hopelessness Impaired self-esteem Impairment of emotional processing Impairment of cognitive functioning Impairment of social functioning No sense of a positive future Physical symptoms [e.g., fatigue, insomnia, lack of energy, etc.] No sense of a positive future Sadness Suicidal ideation Worthlessness
Outcomes	Decreased well-being Decreased global health status Decreased life satisfaction Impaired quality of life Increased mortality Lack of treatment adherence

**Table 5 healthcare-12-00960-t005:** Validity matrix of critical attributes of spiritual distress, hopelessness, and depression.

Factor	Spiritual Distress	Hopelessness	Depression
Characteristics	Lack of meaning and purpose in life	Having given up on life	Multi-impairment Sadness
Death	Existential issues, Frequent thoughts about death	Hastened death	Suicidal ideation
Disconnected to God or the transcendent	Anger towards God or Superior Being. Feel abandoned or punished by God	No	No
Future	Fear of the future	Uncertainty regarding the future	No sense of a positive future
Guilt	Yes	No	Yes
Hope	Hopelessness	Lack of hope	Hopelessness
Life	Lack of meaning and purpose in life	Sense of futility in life	Suicidal ideation
Rupture of belief system Spiritual belief Religious beliefs	Yes	No	No
Self	Disconnected from self Loss or altered sense of self	Inability to improve and control their own situation	Impaired self-esteem

## Data Availability

The data supporting this study’s findings are available in this article’s Appendix A.

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
