# Peer review of "Spiritual Distress, Hopelessness, and Depression in Palliative Care: Simultaneous Concept Analysis"

_healthcare, 2024, doi:10.3390/healthcare12100960_

Round 1
Reviewer 1 Report
Comments and Suggestions for Authors
Summary of the research
The study discusses the importance of distinguishing between spiritual distress, hopelessness, and depression in palliative care. Through a simultaneous concept analysis, it highlights that they are distinct concepts: spiritual distress involves the rupture of spiritual/religious belief systems, hopelessness is a sense of abandonment and lack of control over the situation, while depression is characterized by sadness and multi-faceted impairment. Clarity on these terms is essential for the development of knowledge and intervention strategies in palliative care.
Abstract
The study abstract appears well-structured and offers to the reader a clear idea of the article content. However, we suggest:
· Reduce to 200 words, as per the guidelines for authors of this journal;
· The consideration of both caregivers' and patients' perspectives in the analysis of these concepts needs to be explicitly emphasized (even though both are discussed later in different sections, it's not clear here, it seems to focus only on the patient);
· Ensure clarity and consistency across various sections of the article. In the abstract, it states "The results highlight that spiritual distress, hopelessness, and depression are different concepts," contradicting the conclusions section, which discusses "interrelationships and overlaps among the three concepts."
Keywords
The keywords are adequate regarding the correct number of keywords required by the journal but it is recommended to eliminate the keywords "patients" and those related to the two models, as they are redundant.
Introduction
The introduction section appears generally well-structured and presents the matter of this study, also presenting a good number of cited studies.
· Refine the discussion regarding the roles of caregivers and patients concerning the concepts expressed in these lines (what implications exist for both in palliative care?), if both targets are consistently considered. Clarity is sometimes lacking.
Methods
In this section the authors describe in detail the methodology used to conduct the study and the material used. Overall, this section is well written, but some changes are required:
· Figure 1, section 2. Provide more explicit justification for this choice.
· Figure 1, section 3. Provide more explicit explanation of the "Wilson approach".
· Figure 1, section 4. Clarify further: how was the analysis of the concepts divided? Based on previous expertise on these concepts? By random selection of the group? By choices with alternative motivations?
· Figure 1, section 6. Specify the general timing of the various phases, including phase 6 (did it occur after the development of validity matrices or at different times)?
· Provide more detailed explanation of Figure 1, section 7.
· Figure 1, section 8. Were professionals equally competent in these concepts? Clarify and specify the figures with whom the exposure occurred. Additionally, were there any changes following this phase? Explicitly state both in case of negative and positive outcomes.
Results
In the Results section, the authors present the findings of their research. This part is well written, but we suggest the following modifications:
· Instead of providing a brief summary of each concept (the table serves as a visual summary), use the results section as a real in-depth exploration of the various characteristics, implications, and obstacles to allow for a more comprehensive understanding. It may also be possible to combine Results & Discussion.
· Lines 186-193. Further elaborate on factors between hopelessness and depression?
Discussion
In the Discussion section the authors discuss the emerged data based on inferences and bibliographic references. This part is well written, but we suggest the following modifications:
· “ The data yielded by this study provided clarifying and relevant evidence regarding the concepts of spiritual distress, hopelessness, and depression”. So, for whom? Given the frequent references to caregivers, are these concepts addressed for both caregivers and patients, or just for patients? 1st case: Provide more explicit clarification; if there are differences in how the concepts are applied, they should be mentioned. It could be interesting to highlight them; 2nd case: References to caregivers are redundant and confusing.
· Lines 219-221. How are depression and hopelessness incompatible with the breakdown of the spiritual/religious belief system of palliative patients and caregivers?
· Lines 287-289. It is recommended to expand on considerations regarding future perspectives and implications for interventions.
Conclusion
This section should provide a brief summary of the results and discussion. There are no reviews regarding this section.
References
References and citation style need to be reviewed following the journal's guidelines.
Use of English
The English language used in the paper is adequate to enable the reader to understand the article.
Overview
The article appears interesting and is well-structured in both form and content. However, there are some minor revisions needed to enhance the effectiveness and accuracy of the article.
Author Response
Dear Reviewer,
Thanks for your comments. Please, find the attached file.
Regards

Reviewer 2 Report
Comments and Suggestions for Authors
Thank you for submitting this interesting piece to the journal for review. There appears to be some good work here on a very interesting subject which may be of interest to readers of the journal. However, your piece needs more work before it is ready for publication. Your piece attempts to apply simultaneous concept analysis to terminology within palliative care - spiritual distress, hopelessness and depression. I wonder why you did not address 'suffering'.
The review was completed in April 20022, which was 2 years ago, could this be updated?
The background will need more work, try to guide the reader. It is not entirely clear why you have carried out this work. The focus is on the three concepts in palliative care, I am not sure why there is an over focus on caregivers (not 'careers')? Please review.
I do not agree that palliative care is a relatively new component of health care. The concept of palliative care has been around for many many years, however, the understanding may have developed, please clarify what you mean. As you are writing for an international audience, it is worth remembering that some countries may be much further advanced in applying the principles of palliative care while other countries may still in the early days of applying these principles.
Please do not label people as 'palliative care patients' p2 (91), p3 (102), p11 (296), these are people with palliative care needs. Please consider reviewing this approach within your paper.
There are multiple short paragraphs on page two which could be developed, rather than simply stating points.
Materials and Methods - in figure 1 you move from present to past tense, be consistent. If you are explaining what you did then it should be past tense. Ensure the reader is clear on what you did as a team.
You mention you moved from 260 articles to 80 this needs some further explanation. You do not say anything about the papers/studies, the study approaches, methods, focus, or where they were carried out/written. Please address this. Did you use a quality assessment tool to review these 80 papers for examples CASP or JBI tools?
There is no explanation on how you arrived at your findings and how they related to these 80 papers. You need to explain the process and guide the reader. Some of the material in table 4 needs to be brought into the body of your paper. Once you have explained this, you can address the concepts, guide the reader.
The discussion section will need to be reviewed following the review and editing of the findings section.
The conclusion and implications for practice could be much stronger.
With some more focused work, this paper could be helpful and be published.
I hope this helps.
You use a numbe rof direct quotes and these do not always have a page number.
Comments on the Quality of English Language
There are a number of typos and English corrections that need to be addressed, for example, I think you are referring to 'carers' not 'careers'
Author Response

(The authors gave the same response as above.)
